# Barriers to USMLE Step-1 accommodations: Students with Type 1 Diabetes

**Emily L. Serata**[1]*, **Emily J. Noonan**[2], **Kristina H. Petersen**[3]

1 Cooper Medical School of Rowan University, Camden, NJ, United States of America, 2 University of Louisville School of Medicine, Louisville, KY, United States of America, 3 Department of Biochemistry & Molecular Biology at New York Medical College, Valhalla, New York, United States of America

* serata64@rowan.edu

## Abstract

### Purpose

Students who earn their medical doctorate (MD) in the U.S. must pass the United States Medical Licensing Exam (USMLE) Step-1. The application process for students with disabilities who seek Step-1 accommodations can be arduous, barrier-ridden, and can impose a significant burden that may have long-lasting effects. We sought to understand the experiences of medical students with Type-1 Diabetes (T1D) who applied for Step-1 accommodations.

### Methods

A Qualtrics survey was administered to students enrolled in Liaison Committee on Medical Education (LCME)-accredited MD programs who disclosed having a primary diagnosis of T1D. Basic counts and qualitative inductive analyses were conducted.

### Results

Of the 21 surveys sent, 16 (76.2%) participants responded. Of the 16 respondents, 11 (68.8%) applied for USMLE Step-1 accommodations, whereas 5 (31.2%) did not. Of the 11 who applied for accommodations, 7 (63.6%) received the accommodations requested, while 4 (36.4%) did not. Of those who received the accommodations requested, 5/7 (71.4%) experienced at least one diabetes-related barrier on exam day. Of those who did not apply for Step-1 accommodations, 4/5 (80%) participants reported experiencing at least one diabetes-related barrier on exam day. Overall, 11/16 (68.8%) students experienced barriers on exam day with or without accommodations. Qualitative analysis revealed themes among participants about their experience with the process: frustration, anger, stress, and some areas of general satisfaction.

### Conclusions

This study reports the perceptions of students with T1D about barriers and inequities in the Step-1 accommodations application process. Students with and without accommodations encountered T1D-related obstacles on test day.

**Data Availability Statement:** The data underlying the results presented in the study are available here: https://figshare.com/s/97e926743504b2990045.

**Funding:** Funding of this project was provided by the Alpha Omega Alpha Honor Medical Society Fellow in Leadership program (KHP). The funding will cover a publication fee. Award Number: 0198259 Grant recipient: Kristina H. Petersen Funders did not play any role in the study design, data collection and analysis, decision to publish, or preparation of manuscript. https://www.aamc.org/about-us/aamc-awards/aoa-glaser-distinguished-teacher/2022-petersen.

**Competing interests:** The authors have declared that no competing interests exist.

# Introduction

In the medical educational environment, people with disabilities (PWD) experience many barriers compared to their peers [1, 2]. Accommodations are implemented to provide an equitable opportunity for all individuals to excel and thrive. Title I of the Americans with Disabilities Act (ADA), established in 1990 and amended in 2008, defines an individual with a disability as a person who has a physical or mental impairment that substantially limits one or more major life activities; has a record of such impairment; or is regarded as having such an impairment [3]. Many diagnoses fall into the definition of disability, including mobility and sensory impairments, chronic conditions, and neurodivergent disorders.

In 2023, 5.6% of allopathic medical students disclosed a disability to their institutions and sought accommodations [4]. PWD encounter multiple barriers prior to admission to medical school, and these barriers do not disappear when they matriculate [2]. PWD are often met with structural and attitudinal barriers that may discourage them from disclosing their disability and their associated need for accommodations, thereby preventing them from being able to perform to their full potential [5–8]. While there are still clear gaps in the number of students who identify as having a disability and those who disclose their need for accommodations to the institution [9–11], the number of PWD registering for accommodations has increased [4].

## Barriers to receiving accommodations for the United States Medical Licensing Step-1 Exam

The United States Medical Licensing Step-1 Exam (Step-1) assesses understanding and application of information covered during the didactic years of medical school [12]. Step-1 is typically given as a one-day examination divided into seven 60-minute blocks and administered over 8 hours, with 45 minutes of scattered breaks [12].

Notably, not all PWD apply for Step-1 exam accommodations. The only study that has investigated this issue reported that in the 2018–2019 academic year, 276 students from 73/144 allopathic schools applied for Step-1 accommodations, and of these students, 52% were denied [13]. Of those who took the exam without accommodations after being denied, 32% failed; this Step-1 failure rate is about six times higher than the overall failure rate during the 2018–2019 year [11]! Another study compared the academic performance of medical students with disabilities—with and without accommodations—to those without disabilities. Among students with disabilities who did not have accommodations, Step-1 scores varied by 12.2 points, while scores of those with accommodations differed from controls by only 6 points; these results nearly reached statistical significance [14]. Importantly, both studies presented data about PWD as a group, in aggregate.

Applying to take Step-1 is a multi-step process. First, students pay the fee and apply for a scheduling permit, which allows them to register for the exam. For students who require accommodations, an additional, parallel, and much more arduous process is necessary.

Students must fill out a form [15], which requires the student's identifying and demographic information as well as multiple medical and academic records. Additional documentation is also required from the student's medical school and physician. Once submitted, USMLE policy specifies that they require up to 60 business days to render a decision [16], but many students have reported experiencing longer wait times [17]. Once approved for accommodations, students must call the testing center to schedule their exam; online registration is not available for students with accommodations. If students feel their accommodations are inadequate, they can apply for reconsideration by completing another form [18] with new documentation; however, this process is also subject to a 60-business-day review, which may deter students with school-imposed exam deadlines.

Overall, applying for Step-1 accommodations is a long and difficult process that often results in students obtaining fewer accommodations than they receive at their institution. Based on published data, this can have detrimental effects on PWD [13, 14]. Further investigation is necessary to understand how various students within this heterogeneous group are impacted.

### Type 1 Diabetes and reasonable accommodations

People with varying disabilities require Step-1 accommodations, including those who have a primary diagnosis of T1D (9.5% prevalence in U.S.) [19]. T1D is defined as a chronic autoimmune disease characterized by unstable glucose levels due to insulin deficiency leading to events of hyper/hypoglycemia and is protected under the ADA [20, 21]. Hyperglycemia occurs when glucose levels are high, resulting in extreme thirst, unintentional weight loss, loss of energy, nausea, trouble seeing, confusion, drowsiness, and poor concentration [22]. Conversely, hypoglycemia occurs when glucose levels are low, causing tachycardia, sweating, nervousness, nausea, difficulty concentrating, and confusion [22, 23]. Severe hypoglycemia may cause loss of consciousness, seizure, and coma; such episodes are life-threatening, and require immediate medical attention [23, 24]. Episodes of both hyper- and hypo-glycemia may occur with rapid onset; these incidents are unpredictable and interfere with activities of daily living, employment, and schooling [25]. Therefore, accommodations in the workplace and educational settings are necessary for individuals with T1D to allow them equal opportunities and access. While there are no official, universally-accepted guidelines on testing accommodations for people with T1D, the American Diabetes Association recommends [26]: "breaks to check blood glucose. . . levels, eat a snack, take medication, or go to the bathroom"; "the ability to rest until blood glucose levels become normal"; "the ability to keep diabetes supplies and food nearby"; and "a private area to test blood glucose or administer insulin."

The current study will survey U.S. allopathic medical students with T1D about their experiences applying for Step-1 accommodations. We hope to identify specific barriers faced by students during the application process and explore whether students perceive that their experiences impacted their Step-1 performance and/or their career journey.

### Methods

Between August and September 2023, surveys were sent to students who reported being enrolled in LCME-accredited MD programs and having a primary diagnosis of T1D. These students were recruited in three ways: through listserv emails to learning specialists and disability professionals at allopathic medical schools, via social media posts through the College Diabetes Network, and through the Medical Students with Disability and Chronic Illnesses GroupMe. The survey was administered electronically via Qualtrics to individuals who opted in and qualified to participate. Three follow-up emails were sent to encourage survey responses. This study was deemed exempt by the New York Medical College IRB (20322, 7/17/2023).

### Survey instrument

A survey was developed by the authors (ES and KHP) to understand the experiences of students with T1D applying for Step-1 accommodations. The Qualtrics survey flow varied, as participant answers prompted different questions. Answer modalities included yes/no, one Likert scale (1–7), and many open-ended questions with unlimited word count. The survey's three sections addressed the students' history of T1D and accommodations, the accommodation application process for the Step-1 exam, and their experience on the Step-1 exam day (**S1 Appendix**).

## Data analysis

Basic counts were conducted to determine descriptive statistics. Responses to open-ended questions were analyzed using thematic analysis [27–29]. Authors (ES and KHP) jointly coded these data using Microsoft Excel. These authors revised codes and themes over several cycles. A third author (EJN) reviewed the qualitative data and analysis for validity and reliability. All authors reached consensus on interpretation of qualitative data.

## Positionality and reflexivity

We believe that our standpoints as authors impacts the design, implementation, and analysis of this study [30, 31]. Our author team includes individuals with diverse experiences and identities related to chronic illness/disability. Our team includes people with chronic illness or disability, including T1D, and who are caregivers of people with T1D. These lived experiences are key to this study's central questions and purpose and are reflected in the discussion.

## Results

A total of 21 medical students at LCME-accredited MD programs identified as having T1D and indicated interest in participating in the survey. Of the 21 invited, 16 participants responded (76.2%; hereafter referred to as "participants"; Table 1).

Participants were first separated into two groups: those who applied for USMLE Step-1 accommodations and those who did not. Of the participants who applied for accommodations (N = 11), this group was further divided into those who received the accommodations they requested and those who did not (section 2, survey questions 2 and 4; S1 Appendix).

All 3 participant groups (those who received requested accommodations, those who did not receive requested accommodations, and those who did not apply for accommodations) were then separated into 2 subgroups: those who experienced diabetes-related barriers on exam day and those who did not (survey section 3; question 1; S1 Appendix) (**Fig 1**).

### Participants who applied for USMLE Step-1 accommodations

Of the participants who responded to the survey, 11/16 (68.8%) applied for Step-1 accommodations. Of those who applied, 7/11 (63.6%) participants previously received accommodations

**Table 1. Participant characteristics (percentage); [standard deviation].**

| Characteristics |
| --- |
| **Survey response rate**. . . . . . . .. . . . . . . . . ... . . . . ...16/21 participants (76.2%) |
| **Age at diagnosis with Type 1 Diabetes** |
| 1–15 years. . . . . . . . . . . . . . . . . . . . . . . . . . .... . ..11 participants |
| 15–30 years. . . . . . . . . . . . . . . . . . . . . . . ... .. . ..5 participants |
| **Year of taking USMLE Step-1 Exam** |
| 2021. . . . . . . . . . . . . . . . . . . . . . . . . . . . . . . . . ..4 participants |
| 2022. . . . . . . . . . . . . . . . . . . . . . . . . . . . . . . .. . ..3 participants |
| 2023. . . . . . . . . . . . . . . . . . . . . . . . . . . . . .. . . ..9 participants |
| **Level of stress associated with requesting accommodations (1–7)** |
| Average level of stress. . . . . . . . . . . . . . . . . . . . .4.9 [1.62] |
| **Students who passed USMLE Step-1 on first attempt** |
| Yes . . . . . . . . . . . . . . . . . . . . . . . . . . . . . . . . . . ...15 participants |
| No. . . . . . . . . . . . . . . . . . . . . . . . . . . . . . . . . . . ..1 participants |

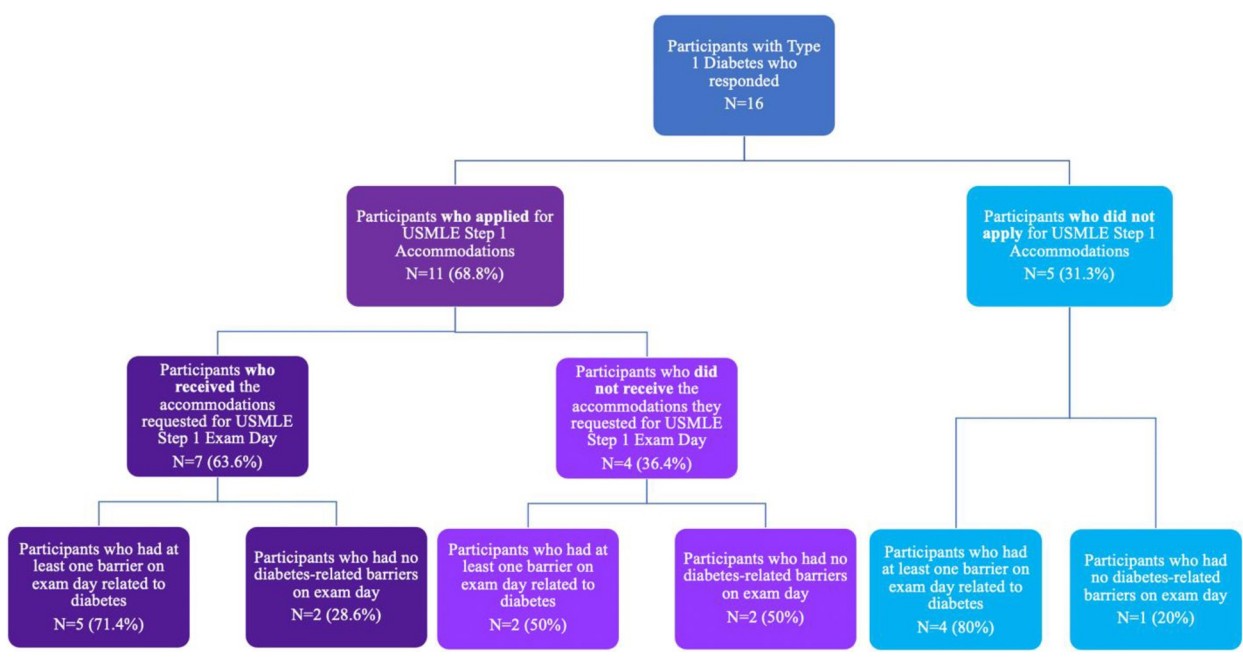

**Fig 1. Data analysis flow chart.**

on standardized exams, with 5/7 (71.4%) specifying that they received accommodations on the MCAT.

The accommodations offered on the application included additional testing time and/or break time increased by 25%, 50%, 100% while also including a space to write additional accommodations [15]. Participants requested a range of accommodations including additional testing time/break time, stop-the-clock time, permission to bring food and water, extended breaks, permission to have and use diabetes management supplies, and a separate room. After applying for accommodations, the NBME contacted 3/11 (27.3%) participants to submit additional documentation, specifically additional letters from endocrinologists more clearly stating the necessity for the accommodations requested.

Of the 11 participants who applied for accommodations, 7/11 (63.6%) received the accommodations they requested, while 4/11 (36.4%) did not. Participants were granted varied accommodation plans, which primarily included extended break times with allowance to bring diabetic supplies into the testing room. The most commonly denied request was additional break-time, which many students requested due to their knowledge that USMLE does not provide stop-the-clock time. Additional accommodation requests that were not approved for some participants included additional variations of extended breaks, extended time, and a separate room. Notably, accommodations varied amongst participants, with some receiving more time than others.

After initial application submission, it took an average of 1.2 months for participants to receive their decision letters, with times ranging from 1 week to 3.25 months. None of the 11 participants chose to go through the appeals process, mainly because they felt that the accommodations received were adequate. However, 2 participants did not choose to go through the process because it was "too long and stressful." On average within the 11 participants who applied, on a scale from 1 to 7, participants rated the level of stress associated with requesting Step-1 USMLE accommodations as 4.9/7 (standard deviation: 1.62).

### Step-1 exam experience: Participants who received requested accommodations

Of the 11 participants who applied for Step-1 accommodations, 7/11 (63.6%) received what they requested. Of these, 5/7 (71.4%) reported experiencing a diabetes-related barrier on exam day. Examples included becoming hypoglycemic during the exam, requiring extra time to gather diabetes supplies from the locker outside of the testing room, and concerns about having snacks and supplies in the room. Of the 7 participants who reported experiencing test-day barriers, 3/7 (42.9%) responded positively, stating that the accommodations granted were "good" and "put them at ease" on exam day.

### Step-1 exam experience: Participants who did <u>not</u> receive requested accommodations

Of the 11 participants who requested Step-1 accommodations, 4 (36.4%) did not receive the accommodations requested. Of these 4 participants, 2 (50%) experienced a diabetes-related barrier on exam day. These barriers included becoming hypoglycemic during the exam and one experience where an insulin pump alarm sounded in the standard testing room and would not stop. This latter event caused the participant to disconnect their insulin pump and leave it outside of the testing room, reportedly resulting in the student experiencing hyperglycemia and difficulty focusing on the exam.

### Participants' feelings about the Step-1 accommodations application process

Participants were asked to share how applying for Step-1 accommodations made them feel. While two respondents reported some areas of general satisfaction, all respondents reported negative emotions surrounding the application process. We identified three primary themes: anger, frustration, and stress. Several students reported more than one feeling; therefore, a given respondent may be represented in more than one category. Responses are cited in Table 2.

### Participants who did <u>not</u> apply for Step-1 accommodations

Of the participants who responded to the survey, 5/16 (31.3%) did not apply for Step-1 accommodations. Four of these participants (80%) reported that they had previously received accommodations on standardized exams; of these, 2 specifically mentioned the MCAT.

Two participants (40%) stated that they did not feel the need to request accommodations and felt that everything they needed (e.g., insulin pump allowed in the exam room) was already approved. Two participants (40%) mentioned that because the Step-1 exam was pass/fail, they did not feel the need to apply for accommodations. One participant stated that "the process for getting accommodations is a months-long ordeal that I did not want to engage in."

Regarding participant experience on exam day, 4/5 (80%) reported experiencing a diabetes-related barrier, including both hypoglycemia (3) and hyperglycemia (1). Most participants were reportedly able to treat these changes, but one participant reported that it "would have been better if I had accommodations. Ran into some time shortage due to blood sugar."

### Additional participant commentary

Participants were given space to include additional comments, which provided nuanced information to contextualize their previous answers. Three themes were identified among

**Table 2. Participants' comments on their feelings about the USMLE accommodations application process.**

| General Satisfaction | • "Overall, I was satisfied with their process for accommodations for Step-1 only because they provided a separate form for health conditions requesting additional break time, specifically mentioning diabetes. The form itself was only 3 pages long and allowed me a free-response section to request juice in the testing room in the event I was stuck on a block for the next hour and my blood sugar dropped."<br>• "I was actually very impressed with how quick the turnaround was after submission of my application." |
|---|---|
| Anger | • "I felt like the system was designed to deny accommodations and to make it as difficult. . .as humanly possible to get through to disincentivize you from applying."<br>• "Made me feel as if a Type 1 diabetic with an insulin pump had never taken Step-1 before."<br>• "Like I had to defend myself for why I deserved accommodations, almost like diabetes isn't 'enough' of a disability to deserve needing extra break time." |
| Frustration | • "I felt like I had to jump through so many hoops to convince people to give me accommodations that I knew I needed. I've had type 1 diabetes for years and a long history of receiving accommodations. It felt unfair that I had to complete such an extensive application that other test takers who didn't need accommodations did not have to worry about, just to ensure that I had an equitable test-taking experience."<br>• "I was a bit paranoid I wouldn't receive accommodations. It was annoying I had to get a letter from my doctor. It wasn't clear what testing supplies I could have or if that required a separate accommodation because I have a CGM (continuous glucose monitor) and insulin pump and that is not listed on their website. You also can't ask for stop the clock breaks. Luckily, my disability services liaison at the medical school had dealt with this situation before and knew you could just request shorter blocks to get more opportunities for break time."<br>• "It honestly was just frustrating. I am still a relatively recent diagnosis and since I had no history of applying for accommodations, that was a hiccup." |
| Stress | • "It was stressful to supply the necessary documentation for such a straightforward diagnosis with what supplies are necessary like a CGM or pump."<br>• "It was additional, unneeded stress."<br>• "It was very worrying and stressful as I felt I would not be able to manage my blood sugar and respond to blood glucose fluctuations optimally. I felt upset that they did not take my request seriously, and worried that I would have a low blood sugar in the testing room during a testing block that I would not be able to treat appropriately. I was also worried that I would have a high blood sugar during the exam and would not have enough time to stop, allow time for my sugar to normalize, and then resume the exam. I worried that my blood sugar fluctuation would impair my mental capacity during the exam."<br>• The application itself was a bit tiresome and long, however. And my school itself took some time to initially process my student verification which delayed everything else and made it more stressful." |

responses: barriers within the process, difficulty accessing information regarding the process, and comments on Step-2CK accommodation applications.

Regarding barriers within the application process, one participant stated that "there should be an alternative way to screen accommodation applicants who have chronic illnesses that do not change in nature. It's exhaust[ing] having to prove that the disease I'm living with requires me to have access to basic things like food, drink, and medication." Another participant stated that "there were a lot of things that it just felt like should have been easier." Both participants acknowledged that they understand that it must be a stringent process to "mitigate any advantage given to examinees;" however they still felt frustration.

In terms of difficulty accessing information about how to apply for accommodations, 3 participants stated that they wished the information on the USMLE website "was clearer" and stated that they had to "reach out to upperclassmen in the medical school" to get further information. A participant also commented that they were "unaware of what types of accommodations to request, what was typically granted to type 1 diabetics, and how to approach the personal statement [part of the application]."

Two participants who did not apply for Step-1 accommodations commented that they did apply for Step-2CK accommodations. One participant stated that the Step-2CK application

process was "understandably tedious" and this participant received approval for accommodations 2 months after submitting. The other participant stated they felt the USMLE "are terrible and illegally withhold accommodations from people." They added that the "USMLE reduced or eliminated all normal accommodations [I] got for [my] medical school exams."

## Discussions/Conclusion

This is the first study to examine barriers faced by medical students with T1D who are in a position to apply for Step-1 accommodations. First, only 11/16 participants reported applying for accommodations. The 5 that chose not to go through the process cited various reasons including feeling they did not need accommodations, not knowing what to seek, and avoiding the "months-long ordeal" associated with applying. Of the 11 who applied, 7/11 received the accommodations requested; however, strikingly 5/7 still experienced a diabetes-related problem on exam day. Furthermore, when we look at all 16 participants (with and without accommodations), 11/16 experienced a diabetes-related barrier on exam day. With respect to their feelings about the application process for Step-1 accommodations, students reported feeling anger, frustration, and stress.

A larger study reported that 52% of PWD who applied for Step-1 accommodations were denied; however individual disability diagnoses were not separated. In our more specific study 36% (4/11) of students with T1D who applied were denied [13]. Our study's small sample size must be taken into consideration as a limitation in interpreting the data; however, even with that in mind, it is possible that this contrast may be attributed to the difference in students who have a disability generally accepted as protected by the ADA (T1D) versus one that is less universally accepted [3].

In our study, many students with T1D experienced barriers in receiving appropriate accommodations for the USMLE Step-1 exam. While there are no officially accepted universal guidelines on testing accommodations for people with T1D, the American Diabetes Association recommends flexible breaks to manage glucose levels while having diabetes supplies nearby [26]. In terms of time accommodations, the USMLE Step-1 offers 25%, 50%, 100% additional test time as well as additional break time [15]; however they do not offer stop-the-clock breaks, which is an accommodation commonly offered on institutional and standardized exams including the SAT, ACT, and MCAT [32, 33]. The AAMC's MCAT website states that stop-the-clock breaks are "granted when time is needed to manage a condition or to utilize a strategy unrelated to your ability to access or process test content" which best meets the needs of students with diabetes, allowing them breaks as needed to manage glucose levels during the exam [32]. Several participants commented on the timing of hypoglycemic episodes being unpredictable—and the necessity for ample time after an episode to "mentally function at [my] best", indicating that stop-the-clock breaks may be necessary for persons with T1D to ensure equal access.

More generally, it is possible that the lack of options and flexibility in accommodations offered by the USMLE may result in students not knowing what types of accommodations to request that best fit their needs, leading to inadequate accommodations, or discouraging them from applying at all. Many students also commented that their difficulty accessing information about accommodations made it difficult to know which available option would be most suitable.

Students enumerated multiple barriers associated with the extensive time investment and delays associated with the application process. Medical students are generally overwhelmed with rigorous curricular obligations, mandated clinical time, expected volunteer hours, research, etc. Students who invest several hours to go through the USMLE accommodations application process must sacrifice time away from pursuing other personal or educational

goals. After awaiting final approval, students are still mandated to call the testing center to schedule their appointment, rather than scheduling with full view of availability online like their peers, adding another barrier.

Many students with diabetes may feel discouraged by the process as they grapple with difficulty in understanding which accommodations are most appropriate to seek and the effort and time it takes to apply and await a decision. With the USMLE Step-1 exam being pass/fail, students may not take the time and effort to apply for accommodations, which may lead to poorer outcomes [13]. Unfortunately, this sentiment was endorsed by participants in our study who stated that they applied for Step-2 accommodations after not applying for Step-1.

While a critique of standardized testing and assessment is beyond the scope of this paper, the barriers experienced by our participants should also be understood as a form of ableism within medical education. Critical Disability Studies scholars use the term "ableism" to describe the "ideas, practices, institutions, and social relations that presume able-bodiedness, and by so doing, construct people with disabilities as marginalized. . ." [34]. Further, the need for accommodation in assessment marginalizes students for whom the standard testing environment is unequitable [35]. As one student reported, "It felt unfair that I had to complete such an extensive application that other test takers who didn't need accommodations did not have to worry about, just to ensure that I had an equitable test-taking experience." Others also used language that highlighted their experiences of marginalization and isolation: "[the process]. . .[m]ade me feel as if a Type 1 diabetic with an insulin pump had never taken Step-1 before." As other scholars in health professions education have argued, ableism adversely impacts both trainees and the practicing workforce [36–38].

Overall, across the three groups of students (**Fig 1**), a strong majority of students (11/16) reportedly experienced diabetes-related barriers on the exam. Shockingly, one participant reported that their insulin pump continued to alarm in the testing room, causing a disruption of the testing environment that resulted in participant having to take off the insulin pump and leave it outside the testing room. This event resulted in the participant not having access to the insulin necessary to manage their diabetes-related symptoms, while also disrupting the additional students in the standard testing room. This example clearly illustrates that proactively providing adequate accommodations for students with diabetes is not only imperative for their well-being, but also to ensure an equitable testing environment for peers taking the exam. A combination of diabetes-related barriers, such as hypo/hyper-glycemia and insulin pump malfunctions, as well as associated stress due to situations like the continual insulin pump alarm seem to both have contributed to poor experiences on test-day. While additional factors, such as stress and anxiety, may have played a role in the negative experiences as has been reported as being the case for many students who take this exam [39–42], participants repeatedly cited diabetes-related barriers as their main source of stress on test-day.

Type 1 Diabetes is commonly known to be a chronic illness that is classified as a disability and protected by the ADA [3]. It is concerning that even for such a commonly known ADA-protected diagnosis, students report experiencing considerable barriers obtaining the accommodations they require to access an equitable testing experience. We speculate that since it is difficult for students with commonly recognized disabilities to obtain accommodations, it is likely more difficult for students with less commonly recognized disabilities to obtain appropriate accommodations. Limitations of this study include the small sample size (N = 16) of participants and the niche nature of the disease. Because the prevalence of Type 1 Diabetes is 5.7% and the amount of medical students with reported chronic disability is significantly decreased from the national prevalence, there was a decreased sample of eligible candidates [43]. We additionally only surveyed those who were MD students at LCME-accredited institutions who had taken USMLE Step 1 within the last 3 years, which was another barrier to increasing our

sample size. While we limited this timeframe in order to decrease the gap in time since students took the exam, results should be viewed through a cautious lens as recall bias could impact our results. One point to consider is that only one individual in the sample did not pass the USMLE Step 1 Exam, potentially decreasing the effect of negative recall bias in most participants. Additional limitations include our selection of students via relevant listservs and networking rather than a randomized approach. Future work includes surveying more individuals with specific disability diagnoses regarding barriers they may face in the USMLE accommodations application process.

As the field of medicine continues to diversify and encourage participation from individuals with varied lived experiences and backgrounds, considering barriers for those with disabilities is critical. While a WHO survey found that 15% of the world's population lives with some form of disability, only 5.6% of medical students in allopathic schools reported having a disability in 2023 [2, 4, 7, 44]. Although a relatively small proportion of medical students have reported disability, the impact of physicians with disabilities on patient care has been studied and shown to be beneficial [2, 45]. Battalova et al. reports, "A shared experience of living with a same or similar disability facilitates a deep sense of understanding that lays a foundation for improved quality of clinician–client interaction" [45]. To continue increasing the number of PWDs pursuing careers in the medical field, we must continue to focus on decreasing potential barriers within the medical education environment.

## Author note

This project was inspired by my personal experience with difficulties in obtaining USMLE Step-1 accommodations. I have had Type 1 Diabetes for more than 10 years now, and throughout my schooling to this point and including the SAT/ACT, MCAT, I received stop-the-clock accommodations with ability to access medications and supplies. It wasn't until I applied for USMLE Step-1 accommodations that I felt I had to prove that my disease was worthy of getting the accommodations I needed to adequately manage my diabetes-related symptoms during the exam. I spent weeks gathering all the required information including doctors' notes, history of accommodations, personal statements, just to apply and be denied after several months of waiting. I appealed and–after several stressful months—was eventually granted extended time. The accommodations I received were not the ones that I had become accustomed to from previous experience (stop-the-clock breaks) and didn't adequately address my needs. While the extended time provided plenty of time to complete the exam, I did have stress that if I became hypoglycemic during a section, I would be unable to take the time I needed to feel better and cognitively fresh. Overall, I felt so frustrated that my peers and I must go through this extensive process to just be given the opportunity to function at everyone else's baseline level. In a time where many students are encouraged to go into medicine with diverse experiences to better understand patients' personal pathologies and experiences in dealing with chronic illness, it feels wrong that those with disability must jump through additional hoops even after being admitted through the same rigorous system as those without disabilities. I wanted to complete this project to draw attention to this issue so that my peers with chronic disability, including but not limited to diabetes, do not have to endure this extra barrier among the several we all deal with every day.

## Supporting information

**S1 Appendix. Survey instrument.**
(DOCX)

## Acknowledgments

Thank you to Suzanne Mosko for inspiring me to complete this project and connecting me to the best mentors.

## Author Contributions

**Conceptualization:** Emily L. Serata, Kristina H. Petersen.

**Data curation:** Emily L. Serata, Emily J. Noonan, Kristina H. Petersen.

**Formal analysis:** Emily L. Serata, Emily J. Noonan, Kristina H. Petersen.

**Funding acquisition:** Kristina H. Petersen.

**Investigation:** Emily L. Serata, Kristina H. Petersen.

**Methodology:** Emily L. Serata, Kristina H. Petersen.

**Project administration:** Emily L. Serata.

**Supervision:** Kristina H. Petersen.

**Writing – original draft:** Emily L. Serata, Kristina H. Petersen.

**Writing – review & editing:** Emily L. Serata, Emily J. Noonan, Kristina H. Petersen.

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
