## [Decision Letter · Decision Letter 0]

6 May 2024

PONE-D-24-10049Barriers to USMLE Step-1 Accommodations: Students with Type 1 DiabetesPLOS ONE

Dear Dr. Serata,

Thank you for submitting your manuscript to PLOS ONE. After careful consideration, we feel that it has merit but does not fully meet PLOS ONE’s publication criteria as it currently stands. Therefore, we invite you to submit a revised version of the manuscript that addresses the points raised during the review process. Please submit your revised manuscript by Jun 20 2024 11:59PM. If you will need more time than this to complete your revisions, please reply to this message or contact the journal office at plosone@plos.org. Please include the following items when submitting your revised manuscript:A rebuttal letter that responds to each point raised by the academic editor and reviewer(s). You should upload this letter as a separate file labeled 'Response to Reviewers'.A marked-up copy of your manuscript that highlights changes made to the original version. You should upload this as a separate file labeled 'Revised Manuscript with Track Changes'.An unmarked version of your revised paper without tracked changes. You should upload this as a separate file labeled 'Manuscript'.If applicable, we recommend that you deposit your laboratory protocols in protocols.io to enhance the reproducibility of your results. Protocols.io assigns your protocol its own identifier (DOI) so that it can be cited independently in the future. For instructions see: https://journals.plos.org/plosone/s/submission-guidelines#loc-laboratory-protocols. Additionally, PLOS ONE offers an option for publishing peer-reviewed Lab Protocol articles, which describe protocols hosted on protocols.io. Read more information on sharing protocols at https://plos.org/protocols?utm_medium=editorial-email&utm_source=authorletters&utm_campaign=protocols.

We look forward to receiving your revised manuscript.

Kind regards,

Mukhtiar Baig, Ph.D.

Academic Editor

PLOS ONE

Journal Requirements:

3. Please upload a copy of Supporting Information Figure/Table/etc. S1 Appendix which you refer to in your text on page 29.

Reviewers' comments:

Reviewer's Responses to Questions

**Comments to the Author**

1. Is the manuscript technically sound, and do the data support the conclusions?

Reviewer #1: Yes

Reviewer #2: Partly

2. Has the statistical analysis been performed appropriately and rigorously? 

Reviewer #1: I Don't Know

Reviewer #2: Yes

3. Have the authors made all data underlying the findings in their manuscript fully available?

Reviewer #1: Yes

Reviewer #2: Yes

4. Is the manuscript presented in an intelligible fashion and written in standard English?

Reviewer #1: Yes

Reviewer #2: Yes

5. Review Comments to the Author

Reviewer #1: Dear researcher,

Thank you for submitting your research. I found your work to be impressive and insightful. However, I noticed that the sample size was relatively small, which might limit the scope of the findings. Additionally, it might be worth considering the possibility of recall bias, as the students who participated in the study were asked to recall their experiences several months after taking the USMLE Step 1 test.

Nevertheless, I appreciate how your research is a commendable effort to identify areas for improvement in the USMLE process to better accommodate students with disabilities. Your work has the potential to make a significant impact on the medical education community, and I encourage you to continue your research in this area.

Thank you for your contributions to the field.

Best regards.

Reviewer #2: The study has tried to explore negative experiences of a marginalized population of medical students (with type 1 DM ) as a result of non-provision of accommodation during USMLE STEP 1 exam. But the sample size is too small, so the results cannot be generalized.

The methodology and statistical analysis are sound for the qualitative design of the study. However, some additional aspects could have been explored like participant's characteristics (e.g. level of stress) that could have contributed to their negative experiences during USMLE exam.

The use of controls in the study would have strengthened the methodology and lead to more meaningful interpretations.

My query to the authors: Is lack of accommodation the only or the major factor for negative experiences of medical students with type 1 DM during USMLE exam? Please elaborate on other factors that could have lead to their ill-health during USMLE exam in the discussion.

6. PLOS authors have the option to publish the peer review history of their article (what does this mean?). If published, this will include your full peer review and any attached files.

Reviewer #1: **Yes: **Ibraheem Algarni

Reviewer #2: No

---

## [Author Response · Author response to Decision Letter 0]

14 May 2024

Reviewer #1: 

Thank you for your time and response. We have added a paragraph in the discussion to address the sample size and recall bias: 

“…Limitations of this study include the small sample size (N=16) of participants and the niche nature of the disease. Because the prevalence of Type 1 Diabetes is 5.7% and the amount of medical students with reported chronic disability is significantly decreased from the national prevalence, there was a decreased sample of eligible candidates[39]. We additionally only surveyed those who were MD students at LCME-accredited institutions who had taken USMLE Step 1 within the last 3 years, which was another barrier to increasing our sample size. While we limited this time-frame in order to decrease the gap in time since students took the exam, results should be viewed through a cautious lens as recall bias could impact our results. One point to consider is that only one individual in the sample did not pass the USMLE Step 1 Exam, potentially decreasing the effect of negative recall bias in most participants…”

Thank you for the continued consideration of this manuscript and please let us know if there are any further changes to be made. 

Reviewer #2: 

Thank you for your time and response. You bring up a good point regarding lack of accommodation being the major factor for negative experiences of medical students with Type 1 DM during the USMLE Exam. We felt that by asking students about their history with past accommodations for exams, that we could better understand their history with accommodations compared to what they received on this exam. While lack of accommodations will not directly cause negative impact on their health during the exam, the lack of accommodations (and anticipation/anxiety over what could occur without accommodations leading up to the exam date) induces stress due to lack of time and space to care for themselves when ill-health outcomes arise. As we describe in our manuscript: 

“…People with varying disabilities require Step-1 accommodations, including those who have a primary diagnosis of T1D (9.5% prevalence in U.S.)[19]. T1D is defined as a chronic autoimmune disease characterized by unstable glucose levels due to insulin deficiency leading to events of hyper/hypoglycemia and is protected under the ADA[20, 21]. Hyperglycemia occurs when glucose levels are high, resulting in extreme thirst, unintentional weight loss, loss of energy, nausea, trouble seeing, confusion, drowsiness, and poor concentration[22]. Conversely, hypoglycemia occurs when glucose levels are low, causing tachycardia, sweating, nervousness, nausea, difficulty concentrating, and confusion[22],[23]. Severe hypoglycemia may cause loss of consciousness, seizure, and coma; such episodes are life-threatening, and require immediate medical attention[23, 24]. Episodes of both hyper- and hypo-glycemia may occur with rapid onset; these incidents are unpredictable and interfere with activities of daily living, employment, and schooling[25]. Therefore, accommodations in the workplace and educational settings are necessary for individuals with T1D to allow them equal opportunities and access…”

Therefore, we feel that accommodations to care for hyper- and hypo- glycemic episodes are necessary for students with T1D to fully demonstrate their abilities. In terms of stress levels, taking this exam is inherently a stressful experience for any student, but most students do not have an additional layer of health concern to consider that could adversely impact their cognition and full ability to perform on the 7-hour exam. All students who had difficulties with their test-taking day specifically described hyper/hypo-glycemic episodes or issues with their diabetes-related technology, such as an insulin pump. 

We stated this information in the results section when citing patients’ specific experiences with adverse events on test-day. We also added 2 sentences addressing this in the discussion:

 “A combination of diabetes-related barriers, such as hypo/hyper-glycemia and insulin pump malfunctions, as well as associated stress due to situations like the continual insulin pump alarm seem to both have contributed to poor experiences on test-day. While additional factors, such as stress and anxiety, may have played a role in the negative experiences as has been reported as being the case for many students who take this exam[39-42], participants repeatedly cited diabetes-related barriers as their main source of stress on test-day…”

Thank you for sharing your concern about the samples size and lack of a control group. 

We also added information to address the small sample size below. We believe the criticism about the lack of a control group is misplaced. The goal of our research was not to study the differences between students with T1D and those without. Rather, our goal was to explore the experiences of students with T1D with the USMLE Step 1 exam. Generally, qualitative research does not include control groups. 

“…Limitations of this study include the small sample size (N=16) of participants and the niche nature of the disease. Because the prevalence of Type 1 Diabetes is 5.7% and the amount of medical students with reported chronic disability is significantly decreased from the national prevalence, there was a decreased sample of eligible candidates[39]. We additionally only surveyed those who were MD students at LCME-accredited institutions who had taken USMLE Step 1 within the last 3 years, which was another barrier to increasing our sample size. While we limited this timeframe in order to decrease the lag since the exam and students’ experiential reports, recall bias still stands as a potential to consider in evaluating our results. One point to consider is that only one individual in the sample did not pass the USMLE Step 1 Exam, potentially decreasing the effect of negative recall bias in most participants. Additional limitations include our selection of students via relevant listservs and networking rather than a randomized approach. Future work includes surveying more individuals with specific disability diagnoses regarding barriers they may face in the USMLE accommodations application process…”

Thank you for the continued consideration of this manuscript and please let us know if there are any further changes to be made.

---

## [Editor Report · Decision Letter 1]

20 May 2024

Barriers to USMLE Step-1 Accommodations: Students with Type 1 Diabetes

PONE-D-24-10049R1

Dear Dr. Serata,

We’re pleased to inform you that your manuscript has been judged scientifically suitable for publication and will be formally accepted for publication once it meets all outstanding technical requirements.

Kind regards,

Mukhtiar Baig, Ph.D.

Academic Editor

PLOS ONE

---

## [Editor Report · Acceptance letter]

27 May 2024

PONE-D-24-10049R1 

PLOS ONE

Dear Dr. Serata, 

I'm pleased to inform you that your manuscript has been deemed suitable for publication in PLOS ONE. Congratulations! Your manuscript is now being handed over to our production team.

Kind regards, 

on behalf of

Professor Mukhtiar Baig 

Academic Editor

PLOS ONE